# Clinical Implication of Preoperative C-Reactive Protein/Albumin Ratio in Malignant Transformation of Intraductal Papillary Mucinous Neoplasm: A Propensity Score Analysis

**DOI:** 10.3390/diagnostics12020554

**Published:** 2022-02-21

**Authors:** Hirotsugu Maruyama, Kojiro Tanoue, Yuki Ishikawa-Kakiya, Masafumi Yamamura, Akira Higashimori, Masaki Ominami, Yuji Nadatani, Shusei Fukunaga, Koji Otani, Shuhei Hosomi, Fumio Tanaka, Noriko Kamata, Yasuaki Nagami, Koichi Taira, Go Ohira, Kenjiro Kimura, Ryosuke Amano, Yasuhiro Fujiwara

**Affiliations:** 1Department of Gastroenterology, Osaka City University Graduate School of Medicine, Osaka 545-8585, Japan; kojitane0622@yahoo.co.jp (K.T.); yukinaze010@hotmail.com (Y.I.-K.); y_masa0520@yahoo.co.jp (M.Y.); higamo@med.osaka-cu.ac.jp (A.H.); komesoudoh@yahoo.co.jp (M.O.); dada@med.osaka-cu.ac.jp (Y.N.); sfukunag0914@yahoo.co.jp (S.F.); kojiotani@med.osaka-cu.ac.jp (K.O.); m1265271@med.osaka-cu.ac.jp (S.H.); m2079981@med.osaka-cu.ac.jp (F.T.); nkamata@med.osaka-cu.ac.jp (N.K.); yasuaki-75@med.osaka-cu.ac.jp (Y.N.); koichit0802@gmail.com (K.T.); yasu@med.osaka-cu.ac.jp (Y.F.); 2Department of Surgical Oncology, Osaka City University Graduate School of Medicine, Osaka 545-8585, Japan; m1153123@med.osaka-cu.ac.jp (G.O.); kenjirokimura@hotmail.com (K.K.); ramano@med.osaka-cu.ac.jp (R.A.)

**Keywords:** intraductal papillary mucinous neoplasm, inflammation-based score, C-reactive protein albumin ratio, malignant transformation

## Abstract

Background: Inflammation-based scoring has been reported to be useful for predicting the recurrence and prognosis of various carcinomas. This study retrospectively investigated the relationship between inflammation-based score and intraductal papillary mucinous neoplasms (IPMNs). Methods: Between January 2013 and October 2018, we enrolled 417 consecutive patients with pancreatic tumors who received surgical resections at our hospital. The main outcome was the association between the preoperative inflammation-based score and their accuracy in predicting malignant transformation of IPMN. Results: Seventy six patients were eligible. Pathological findings indicated that 35 patients had low-grade dysplasia, 18 had high-grade dysplasia, and 23 had invasive carcinomas. As the C-reactive protein albumin ratio (CAR) was higher, malignant transformation of IPMNs also increased (*p* = 0.007). In comparing CAR^high^ and CAR^low^ using cutoff value, the results using a propensity score analysis showed that the CAR^high^ group predicted malignant transformation of IPMNs (odds ratio, 4.18; 95% confidence interval, 1.37–12.8; *p* = 0.01). In the CAR^high^ group, disease-free survival (DFS) was significantly shorter (*p* = 0.04). In the worrisome features, the AUC for the accuracy of malignant transformation with CAR^high^ was 0.84 when combining with the MPD findings. Conclusions: Preoperative CAR could be a predictive marker of malignant transformation of IPMNs.

## 1. Introduction

Intraductal papillary mucinous neoplasms (IPMNs) are the most frequent pancreatic cystic neoplasms with malignant potential [1,2]. Recently, there has been an increase in IPMN diagnoses due to the enhanced accuracy of images and recognition of the disease [3]. However, the inability to determine the benign or malignant nature of IPMN based on image findings creates a clinical problem.

In 2017, the IPMN International Consensus Guidelines were revised [4]. The factors of high-risk stigmata (HRS) and worrisome features (WF) were changed, and a “mural nodule (MN) size of 5 mm or more” was specified. Additionally, according to the guidelines, the indications for surgery are HRS (obstructive jaundice, enhancing MN ≥ 5 mm, main pancreatic duct (MPD) ≥ 10 mm). However, postoperative histopathological findings often show adenoma rather than cancer, and the diagnostic disagreements between preoperative imaging findings and postoperative histopathological findings are problematic [5,6,7].

Clinically, IPMNs tend to be over-treated with surgery. On the other hand, invasive cancer has a poor prognosis, and it is important to carefully judge the indication for surgery. Furthermore, there are cases where cancer is present and cases where the transition to invasive examination such as endoscopic retrograde cholangiopancreatography (ERCP) is needed in the WF [8,9]. These cases need very careful follow-up. At present, image findings and tumor markers alone have limitations. Therefore, novel biomarkers with simple and higher accuracy for predicting the presence of malignancy are needed.

Systemic inflammatory reactions are used as biomarkers, as they play a crucial role in the malignant transformation and progression of various solid tumors [10,11,12,13,14]. The existence of systemic inflammation, as measured by parameters such as neutrophil-to-lymphocyte ratio (NLR), Glasgow Prognostic Score (GPS), C-reactive protein albumin ratio (CAR), and lymphocyte-to-monocyte ratio (LMR), are associated with poor prognosis across multiple malignancies, including pancreatic cancer [15,16]. Recently, NLR and PLR have been reported to be predictive markers for the presence of invasive carcinoma in pancreatic cysts [13,17]. However, the relationship with inflammation-based scores in the patients with IPMNs has rarely been reported [17,18,19]. Therefore, we hypothesized that preoperative values of the preoperative inflammation-based scores are predictive factors for malignant transformation of IPMNs.

The aim of study was to investigate the relationship between preoperative the preoperative inflammation-based scores and the malignant transformation of IPMNs.

## 2. Materials and Methods

### 2.1. Patients

This research was a retrospective cohort study performed at a single referral hospital. Between January 2013 and October 2018, 417 consecutive patients with pancreatic tumors underwent surgical resection at the Department of Surgical Oncology, Osaka City University Graduate School of Medicine. In addition, clinical information on postoperative histopathological findings was collected retrospectively from electronic medical records. The inclusion criteria were IPMN, low-grade dysplasia (LGD), high-grade dysplasia (HGD), or invasive carcinoma (INV). The exclusion criteria were as follows: (i) pancreatic ductal adenocarcinoma (PDAC) (including IPMN with PDAC), (ii) bile duct cancer, (iii) neuroendocrine tumor, (iv) solid-pseudopapillary neoplasm, (v) metastatic pancreatic tumor, (vi) mucinous cystic neoplasm, (vii) serous cystic neoplasm, (viii) benign disease (pseudo pancreatic cyst, chronic pancreatitis, and pancreaticobiliary maljunction).

### 2.2. Ethical Considerations

The ethics committee of the Osaka City University Graduate School of Medicine approved the study’s protocol (number 4342). In addition, we provided all patients with the opportunity to opt out of the study on our website’s home page.

### 2.3. Main Outcome Measurements

We investigated whether the preoperative value of the preoperative inflammation-based scores is useful for predicting malignant transformation of IPMN.

### 2.4. Data Collection

Clinical patient data were collected as follows: age, sex, IPMN type (Main duct type, Mixed type, Branch duct type), number of cysts (Unifocal, Multifocal), operation procedure, tumor location, size of cyst, MPD diameter, presence of MN, presence of jaundice, and history of pancreatitis. Preoperative neutrophil count, lymphocyte count, platelet count, monocyte count, high-sensitivity C-reactive protein (CRP), albumin, jaundice, carcinoembryonic antigen (CEA), and carbohydrate antigen 19-9 (CA19-9) were collected from the blood samples within seven days before surgery. The inflammation-based scores, including NLR, CAR, LMR were examined using those samples. We defined the normal range of CEA and CA19-9 as 0 to 5 ng/mL and 0 to 37 U/mL, respectively, at our hospital.

### 2.5. Histopathological Assessment

The resected specimens were evaluated according to criteria defined by a recent consensus. The histological grade of IPMN was evaluated as LGD, HGD, or INV [20].

### 2.6. Imaging Assessment

We evaluated IPMN type, number of cysts, tumor location, size of cyst, MPD diameter, and presence of MN based on available imaging data from endoscopic ultrasonography (EUS) and/or computed tomography (CT)/magnetic resonance imaging (MRI).

### 2.7. Endoscopic Procedure

We used echoendoscope (GF-UCT260, GF-UCT240, and GF-UE260; Olympus Medical System, Tokyo, Japan) and ultrasound processer (Aloka ProSound α5 and F75) under conscious sedation for all patients. Both trainees and experts performed EUS, because our hospital is a teaching hospital. Trainees were assisted by experts as needed to ensure procedural quality when performing EUS.

### 2.8. Imaging Acquisition

Contrast-enhanced CT examination was performed with multi-detector CT machines. Arterial phase scanning began 35–40 s after injection of 2 mL/kg of body weight of a nonionic iodinated contrast agent at a rate of 4 mL/s with a bolus-triggered technique using an automatic power injector. Portal and delayed phase scanning were begun 70 and 180 s after the start of the contrast medium injection, respectively. The slice thickness was 2 mm or 5 mm.

The MRI examination was performed using a 3.0 Tesla system (Ingenia; Philips Healthcare, Best, The Netherlands). MRI images were acquired using the following sequences: a T1 weighted sequence (in-phase and out-of-phase), T2 weighted sequence, FAT-SAT sequence, diffusion-weighted sequence, magnetic resonance cholangiopancreatography.

### 2.9. Statistical Analysis

To summarize baseline patients’ clinical and demographical characteristics, medians and interquartile ranges were used for continuous variables and percentages, and counts were used for categorical variables. Regarding the categorical variables, comparisons were performed using the chi-squared test or Fisher’s exact test when necessary, because of the small sample sizes. For the continuous variables, comparisons were performed using Student’s *t*-test. The Jonckheere-Terpstra test was used to analyze the tendency of the systemic inflammation markers and three groups (LGD, HGD and INV). The diagnostic accuracy, including sensitivity, specificity, and accuracy, was calculated for each parameter using the cutoff values determined by the receiver operating characteristic (ROC) curve analysis. The cutoff values were defined as the highest sensitivity and specificity that lay closest to the left upper corner of the ROC curve. First, binary logistic regression analysis was used to evaluate the predictive factors for the presence of HGD and INV, estimated by calculating the odds ratios (ORs) and the 95% confidence intervals (CIs). Secondly, to evaluate the association between the CAR and IPMN with LGD and HGD/INV, the multivariable regression model was used with adjustments for age, sex, presence of MN, MPD ≥ 10 mm, and the presence of jaundice. In this model, a nonlinear restricted cubic spline was drawn to allow a nonlinear determination for normalization of the CAR on IPMN with LGD and HGD/INV along with adjustment for the set of covariates described above. All statistical inferences were performed using two-sided testing at the 5% significance level. All statistical analyses were conducted with R software version 3.6.1.

In addition, the inverse probability of treatment weighting (IPTW) method was applied to assess the predictive factors of IPMN with HGD/INV. This method can adjust for confounding factors and evaluate causal effects without reducing sample size using the estimated propensity scores to assign weights to the data. Statistical analyses were performed using IBM SPSS software, version 23.0 for Windows (IBM Corp., Armonk, NY, USA).

## 3. Results

### 3.1. Baseline Characteristics of Patients

A total 76 patients with histopathological diagnosed IPMN were included in this study (Figure 1). Patient clinicopathological characteristics of this study subject are shown in Table 1. The median age was 73 years (range 69–76), the proportion of males was 61%, and the median CAR was 0.02 (range 0.006–0.09). Regarding histological grade, 35 patients (46%) had LGD, 18 (23.6%) had HGD, and 23 (30.4%) had invasive INV. Some patients had a history of other organ cancer; however, there were no recurrences, and none were receiving chemotherapy or radiation therapy. In the comparison between the LGD group and the HGD / INV group, WF was significantly higher in the LGD group (*p* = 0.01), and CAR tended to be higher in the HGD/INV group (*p* = 0.06). CA19-9 showed a significant difference; however, most cases were within the normal range.

### 3.2. Preoperative Inflammation-Based Scores

We focused on each marker to differentiate IPMN with HGD and INV from IPMN with LGD in terms of the surgical indication. Figure 2 shows that the CAR of IPMN with HGD and INV were higher than that of IPMN with LGD (*p* = 0.007). No significant difference was found between NLR, PLR, and LMR. In addition, the higher the grade (HGD and INV) in the multivariable regression model with restricted cubic spline, the higher the CAR, and it was positively correlated (*p* = 0.047). The analysis showed that a higher CAR would increase the malignant transformation of IPMN. According to this result, we focused on CAR and calculated the cutoff value from the ROC curve as 0.011 (Figure 3).

### 3.3. Clinical Outcome

We divided the patients into two groups, CAR^low^ and CAR^high^, according to the optimal cutoff value of CAR (0.011). Compared with patients of the CAR^low^ group, patients of CAR^high^ group were no significant differences in age, sex, IPMN type, cyst size, MPD diameter, MN, and LMR. However, CEA and PLR were significantly higher in the CAR^low^ group (Table 2). Univariate and multivariate analyses for predictive factors of IPMN with HGD/INV are shown in Table 3. Predictive factors of IPMN with HGD/INV in the univariate analysis was CAR^high^ (OR, 3.28; 95% confidence interval [CI], 1.24–8.69; *p* = 0.02) and ≥5 mm contrast MN (OR, 6.18; 95% CI, 2.11–18.1; *p* < 0.01). Similar results were indicated in multivariate analysis (CAR^high^; OR, 3.84; 95% CI, 1.28–11.5; *p* = 0.02, ≥5 mm contrast MN; OR, 5.38; 95% CI, 1.71–16.9; *p* < 0.01) (Table 3).

In addition, after adjustment for confounding factors using IPTW, CAR^high^ was also predictive of IPMN with HGD/INV (OR, 4.18; 95% CI, 1.37–16.8; *p* = 0.01) (Table 4).

### 3.4. Association of the CAR with Overall Survival and Disease-Free Survival

The median follow-up period for all patients was 879.5 days (95% CI 503.8–1232.8 days); disease recurred in 11 (14.5%) patients, and 8 (10.5%) patients died.

In comparisons of CAR^high^ and CAR^low^, no significant difference was shown in OS (CAR^high^ vs. CAR^low^; 1733.5 vs. 1771.3 days, *p* = 0.89). However, CAR^high^ was significantly shorter in DFS (1550.7 vs. 1902.8 days, *p* = 0.04) (Figure 4).

### 3.5. Diagnostic Accuracy of CAR for WF Patients with HGD/INV

We investigated the accuracy of HGD/INV diagnosis for each item of CAR^high^ and WF for 23 WF patients using ROC curve analysis. LGD had 16 patients (69.6%), and HGD/INV had 7 patients (30.4%). The diagnostic accuracy of CAR^high^ and each item was as follows: CAR^high^, AUC (0.64; 95% CI, 0.42–0.86); MPD 5–9 mm, AUC (0.78; 95% CI, 0.66–0.91); 5 mm/2 years increase, AUC (0.56; 95% CI, 0.48–0.65); lymph node enlargement, AUC (0.5; 95% CI, 0.50–0.50); wall thickening, AUC (0.65; 95% CI, 0.44–0.87); 5 mm > nodule, AUC (0.55; 95% CI, 0.34–0.76); 30 mm < cyst, AUC (0.69; 95% CI, 0.47–0.91). The diagnostic accuracy of CAR^high^ was not high, but it was improved by combining it with MPD 5–9 mm, AUC (0.84; 95% CI, 0.70–0.99) (Figure 5).

## 4. Discussion

We investigated the relationship between the preoperative inflammation-based scores and the malignant transformation of IPMN and found that the CAR^high^ (≥0.011) is a predictive factor for malignant transformation of IPMN. In addition, the analysis showed that a higher value of CAR positively correlates with the malignant transformation of IPMN. These results show that CAR can be used as a simple and clinically meaningful biomarker for the malignant transformation of IPMN.

It is widely known that host–tumor interaction between local cancer and an individual has a significant effect on their general condition, including cancer patients’ nutritional status and immunocompetence. Patients with cancer often have systemic inflammation, and a combination of serum CRP and albumin level is used as a method for evaluating inflammatory response and nutritional status. McMillan et al. reported GPS to be an excellent prognostic marker for non-small cell lung cancer [21].

Many studies have also reported that GPS plays a pivotal role in the progression of various malignancies, and is closely related to poor prognosis in patients [22,23,24,25,26]. In recent years, CAR has also been reported to have clinical implications and usefulness in many studies, including research regarding pancreatic cancer [27,28,29,30,31]. Cancer patients have chronically increased Interleukin-6 (IL-6) in the circulating blood, affecting the acute-phase proteins (APPs); positive APP corresponds to CRP, and negative APP to albumin. Therefore, it is an index that indirectly indicates the presence of systemic metabolic abnormality derived from circulating IL-6 in cancer patients, independent of the clinical stage of cancer.

In addition, the association between NLR and invasive carcinoma of IPMN has been reported [17,18,19]. However, these studies did not related to predictive markers for the presence of cancer, including HGD. IPMN should be resected before progression to invasive carcinoma, because IPMN with associated invasive carcinomas has significantly poorer prognosis than that without invasion. We found CAR to be a predictive marker for the presence of cancer, including HGD, in IPMN patients. In the present study, we focused on patients with respectable IPMN rather than advanced cancer. Therefore, we performed predictions for those with mild systemic inflammation and good nutritional status, using CAR as the basis for evaluation. The median CAR was 0.02 (range 0.006–0.09), and there were no GPS2. The preoperative value of CAR (≥0.011) was an independent predictive factor for the presence of HGD/INV in the univariate and multivariate analysis, and a higher CAR value further increased this likelihood.

Previous studies have reported that higher inflammatory markers, including CAR, are associated with higher histological grades from LGD to HGD [17,18]. In these results, a minor host response of intraepithelial neoplastic changes leading to peripancreatic inflammatory cell infiltration and increased intracystic inflammation with cytokine production such as IL and prostaglandin are related. This suggests that systemic inflammation differs not only in INV, but also between LGD and HGD, supporting the present study.

In the present study, CAR^high^ was correlated with significantly shorter DFS. The outcomes of patients with cancer were determined with respect to patient-related factors, and the presence of preoperative systemic response in patients have been reported to predict poor survival after resection of several gastrointestinal cancers [25,32,33]. In particular, the predictive value of the CAR has been reported to be superior to mGPS [27]. CAR was able to reflect the potential inflammatory state, and is believed to be able to lead to the prediction of DFS. In addition, the potential inflammatory state could be reflected in patients with WF. In our results, the diagnostic accuracy of CAR^high^ was not high, but it was improved by combining it with MPD 5–9 mm.

The clinical implications of these study findings are that CAR can be used as a simple and objective biomarker for malignant transformation in patients with IPMN. Previous reports of NLR and PLR were able to predict INV, but not HGD. However, CAR was able to predict HGD and INV. We believe that the use of high-sensitivity CRP reflects minor changes that do not affect NLR or PLR. Therefore, high values of CAR will help predict the presence of cancer, including HGD, and transition to invasive examination such as ERCP and EUS. In addition, our results suggest that it could predict recurrence a short period of time before surgery. Since these results are able to predict the prognosis of patients with IPMN, we believe that they will contribute to study for improving prognosis in the future.

The major strength of the present study is that it is the first report describing the relationship between the preoperative value of CAR and the malignant transformation of IPMN. IPMN is difficult to diagnose in benign or malignancy without pathological results on image findings. EUS is the best inspectional method; however, the accuracy depends on the skill of the endoscopist. In addition, the evidence for the revisions in the IPMN International Consensus Guidelines 2017 is still unsatisfactory. In these situations, it is significant a relationship has been found with a biomarker capable of calculating the same value.

The present study has several limitations. First, this is based on retrospectively collected data in a single center. Second, this is a small sample. Therefore, this may cause the statistical power to be low, and selection bias to be present. Therefore, we used IPTW analysis. This method can adjust for confounding factors and evaluate causal effects without reducing the sample size using the estimated propensity scores to assign weights to the data. All included patients were histopathologically diagnosed with IPMN. However, we did not investigate the patients with IPMN without indication for resections. Third, some patients did not undergo EUS. EUS has been proven to be superior to CT/MRI for detecting pancreatic lesions. Therefore, preoperative evaluation of the mural nodules might be inadequate. However, in the revised IPMN International Clinical Practice Guidelines 2017, contrast-enhanced EUS is described as the best for the evaluation of mural nodules, but CT/MRI is also acceptable, and clinical error is small. In the future, multicenter prospective studies with larger numbers of patients that include systemic inflammatory markers are needed to confirm the study results. In the results, systemic inflammatory markers might provide a possible clue for estimating the malignant transformation of IPMN.

## 5. Conclusions

The preoperative value of CAR is a predictive factor for the malignant transformation of IPMN, which increases with higher CAR. CAR can be used as a simple and clinically meaningful biomarker for malignant transformation of IPMN.

## Figures and Tables

**Figure 1 diagnostics-12-00554-f001:**
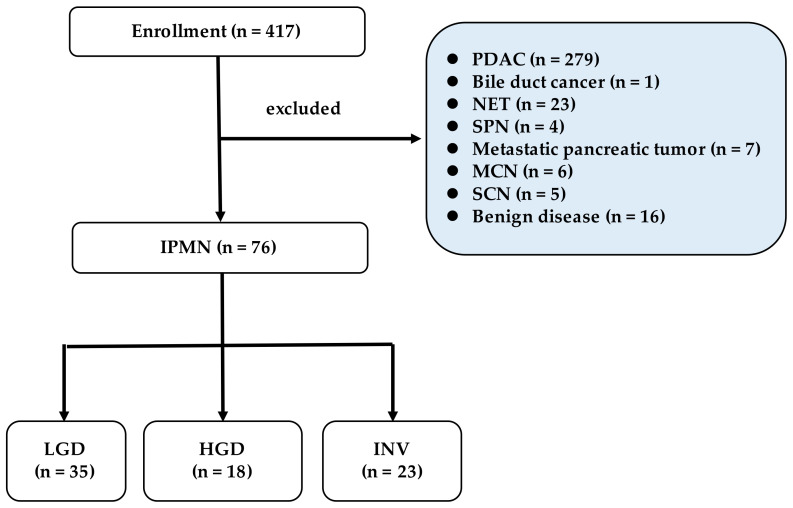
Diagram illustrating the study design. PDAC: pancreatic ductal adenocarcinoma, NET: neuroendocrine tumor, SPN: solid-pseudopapillary neoplasm, MCN: mucinous cystic neoplasm, SCN: serous cystic neoplasm, IPMN: intraductal mucinous neoplasm, LGD: low-grade dysplasia, HGD: high-grade dysplasia, INV: invasive carcinoma.

**Figure 2 diagnostics-12-00554-f002:**
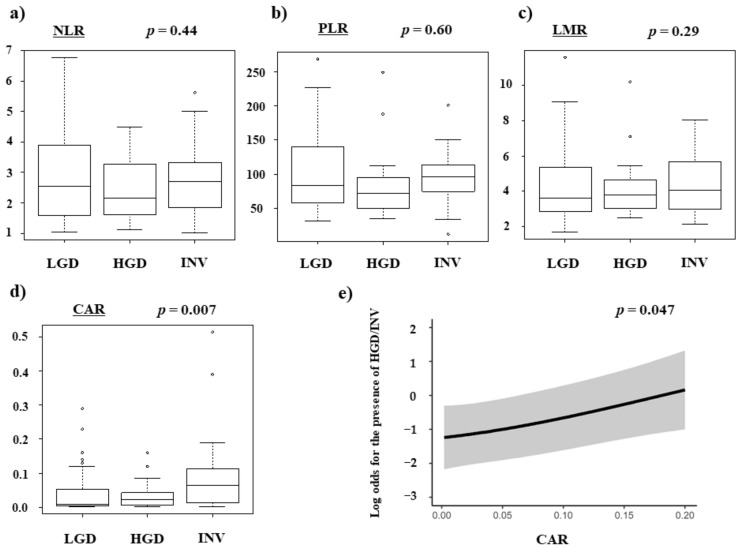
The association between the preoperative inflammation-based scores and 3 subgroups (LDG, HGD and INV). (**a**) NLR. (**b**) PLR. (**c**) LMR. (**d**) CAR. (**e**) The analysis was positively correlated (*p* = 0.047) and showed that a higher CAR would increase the malignant transformation of IPMN. IPMN: intraductal mucinous neoplasm. NLR: neutrophil-to-lymphocyte ratio, PLR: platelet-lymphocyte ratio, LMR: lymphocyte-to-monocyte ratio, CAR: C-reactive protein albumin ratio, LGD: low grade dysplasia, HGD: high grade dysplasia, INV: invasive carcinoma.

**Figure 3 diagnostics-12-00554-f003:**
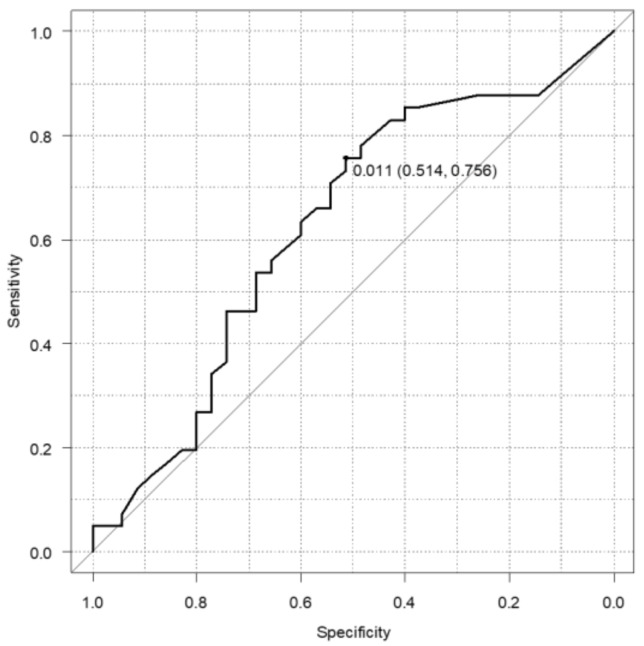
ROC curve of CAR. The cutoff was calculated 0.011 from the ROC curve. ROC: receiver operating characteristic, CAR: C-reactive protein albumin ratio.

**Figure 4 diagnostics-12-00554-f004:**
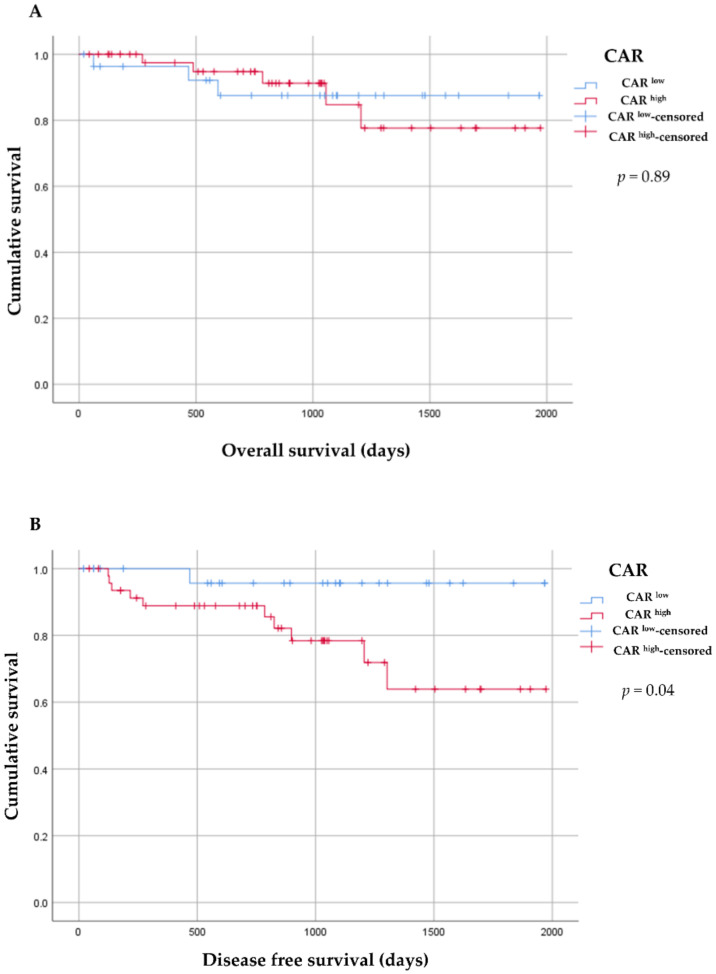
Kaplan–Meier survival curves. (**A**) Overall survival (OS). (**B**) Disease free survival (DFS). Kaplan–Meier survival curves showing the difference between the CAR^high^ and CAR^low^ groups in DFS. OS showed no significant difference; however, CAR^high^ group was significantly shorter in DFS.

**Figure 5 diagnostics-12-00554-f005:**
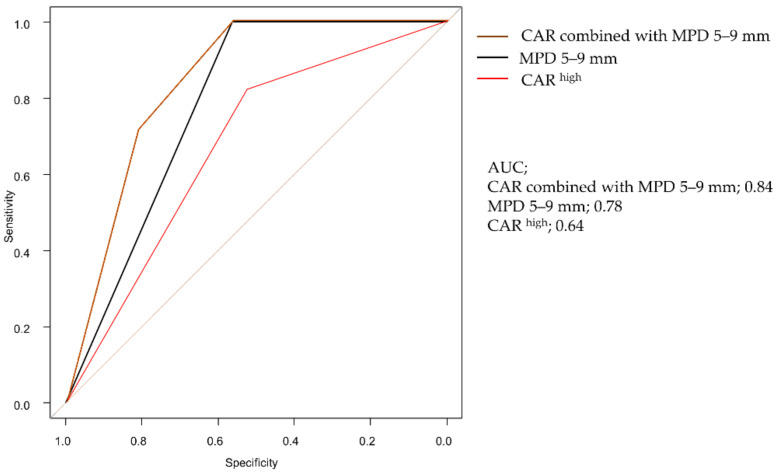
Receiver operating characteristics (ROC) curve for CAR^high^ combined with Main pancreatic duct (MPD) 5–9 mm, MPD 5–9 mm and CAR^high^. CAR^high^ combined with MPD 5–9 mm is 0.84.

**Table 1 diagnostics-12-00554-t001:** Patient clinicopathological characteristics.

		Total	LGD	HGD and INV	*p*-Value
Patients, n. (%)		76 (100)	35 (46)	41 (54)	
Age, median (IQR), years		73 (69–76)	73 (68.5–75.5)	73 (71.0–76.0)	0.71
Sex, n. (%)	Male	46 (61)	23 (65.7)	23 (56.1)	0.48
	Female	30 (39)	12 (34.3)	18 (43.9)	
IPMN type, n. (%)	Main duct type	8 (10.5)	2 (5.7)	6 (14.7)	0.48
	Mixed type	32 (42.1)	16 (45.7)	16 (39.0)	
	Branch duct type	36 (47.4)	17 (48.6)	19 (46.3)	
Location, n. (%)	Head	45 (59)	24 (68.6)	21 (51.2)	0.16
	Body or tail	31 (41)	11 (31.4)	20 (49.8)	
Number of cyst lesion, n. (%)	Unifocal	54 (71.1)	24 (68.5)	30 (73.2)	0.79
	Multifocal	22 (28.9)	11 (31.5)	11 (26.8)	
Cyst size (mm), median (IQR)		22.9 (16.9–32.6)	26.0 (20.5–31.5)	21.7 (14.7–37.6)	0.21
MPD diameter (mm), median (IQR)		5.45 (3.8–8.35)	5.5 (3.4–8.4)	5.4 (3.9–8.3)	0.76
Mural nodule (mm), median (IQR)		5.65 (0–8.58)	4.9 (0–7.4)	6.2 (0–11.6)	0.08
Jaundice, n. (%)		5 (6.6)	1 (2.9)	4 (9.8)	0.37
History of pancreatitis, n. (%)		9 (11.8)	2 (5.7)	7 (17)	0.17
Worrisome features, n. (%)		23 (30.2)	16 (45.7)	7 (17)	0.01
High-risk stigmata, n. (%)		48 (63.2)	18 (51.4)	30 (73.2)	0.06
CEA, ng/mL, median (IQR)		3.2 (1.9–5.3)	3.0 (1.7–4.75)	3.3 (2.3–5.3)	0.48
CA19-9, IU/mL, median (IQR)		8 (5–19.5)	5.0 (4.0–13)	13 (7.0–24)	0.02
NLR, median (IQR)		2.3 (1.6–3.72)	2.54 (1.58–3.90)	2.32 (1.66–3.27)	0.87
LMR, median (IQR)		3.9 (2.7–5.1)	3.61 (2.85–5.36)	3.86 (3.04–5.03)	0.67
PLR, median (IQR)		84.5 (59.3–123.8)	83 (58.5–140)	86 (65–107)	0.56
CAR, median (IQR)		0.02 (0.006–0.09)	0.01 (0.005–0.05)	0.03 (0.01–0.09)	0.06
Operation procedure, n. (%)	PD	40 (52.6)	20 (57.1)	20 (48.7)	
	DP	31 (40.8)	14 (40)	17 (41.5)	
	TP	3 (3.9)	1 (2.9)	2 (4.9)	
	MP	2 (2.7)	0	2 (4.9)	

IQR: interquartile range, IPMN: intraductal mucinous neoplasm, MPD: main pancreatic duct, CEA: carcinoembryonic antigen, CA19-9: carbohydrate antigen, NLR: neutrophil-to-lymphocyte ratio, LMR: lymphocyte-to-monocyte ratio, PLR: platelet-lymphocyte ratio, CAR: C-reactive protein albumin ratio, PD: pancreaticoduodenectomy, DP: distal pancreatectomy, TP: total pancreatectomy MP: middle pancreatectomy, LGD: low grade dysplasia, HGD: high grade dysplasia, INV: invasive carcinoma.

**Table 2 diagnostics-12-00554-t002:** Comparison of clinical characteristics between CAR^low^ and CAR^high^.

		Total	CAR^low^	CAR^high^	*p*-Value
Patients, n. (%)		76	28 (36.8)	48 (63.2)	
Age, median (IQR), years		73 (69–76)	73 (65.5–76.0)	73 (69–75.3)	0.74
Sex, n. (%)	Male	46 (60.5)	16 (57.1)	30 (62.5)	0.81
	Female	30 (39.5)	12 (42.9)	18 (37.5)	
IPMN type, n. (%)	Branch duct type	40 (52.6)	13 (46.4)	27 (56.2)	0.48
	Mixed, Main duct type	36 (47.4)	15 (53.6)	11 (43.8)	
Location, n. (%)	Head	45 (59.2)	15 (53.6)	30 (62.5)	0.48
	Body or tail	31 (40.8)	13 (46.4)	18 (37.5)	
Number of cyst lesion, n. (%)	Unifocal	54 (71.1)	18 (64.3)	36 (75)	0.4
	Multifocal	22 (28.9)	10 (35.7)	12 (25)	
Cyst size (mm), n. (%)	<30	52 (68.4)	20 (71.4)	32 (66.7)	0.8
	≥30	24 (31.6)	8 (28.6)	16 (33.3)	
MPD diameter (mm), n. (%)	<10	59 (77.6)	24 (85.7)	35 (72.9)	0.26
	≥10	17 (22.4)	4 (14.3)	13 (27.1)	
Mural nodule, median (IQR), mm		5.65 (0–8.78)	5.85 (0–7.78)	5.45 (0–9.45)	0.86
≥5 mm contrast mural nodule, n. (%)	Present	29 (38.2)	9 (32.1)	20 (41.7)	0.47
Jaundice, n. (%)	yes	5 (6.6)	0 (0)	5 (10.4)	0.15
History of pancreatitis, n. (%)	yes	9 (11.8)	3 (10.7)	6 (12.5)	1
Worrisome features, n. (%)	yes	23 (30.3)	11 39.3)	12 (25)	0.21
High-risk stigmata, n. (%)	yes	48 (63.2)	17 (60.7)	31 (64.6)	0.81
CEA, ng/mL, n. (%)	≤5	55 (72.4)	25 (89.3)	30 (62.5)	0.02
	>5	21 (27.6)	3 (10.7)	18 (37.5)	
CA19-9, IU/mL, n. (%)	≤37	65 (85.5)	25 (89.3)	40 (83.3)	0.74
	>37	11 (14.5)	3 (10.7)	8 (16.7)	
NLR, n. (%)	<3.27	51 (67.1)	22 (78.6)	29 (60.4)	0.13
	≥3.27	25 (32.9)	6 (21.4)	19 (39.6)	
LMR, n. (%)	<2.64	11 (14.5)	3 (10.7)	8 (16.7)	0.74
	≥2.64	65 (85.5)	25 (89.3)	40 (83.3)	
PLR, n. (%)	<107	50 (65.8)	23 (82.1)	27 (56.3)	0.03
	≥107	26 (34.2)	5 (17.9)	21 (43.7)	

IQR: interquartile range, IPMN: intraductal mucinous neoplasm, MPD: main pancreatic duct, CEA: carcinoembryonic antigen, CA19-9: carbohydrate antigen, NLR: neutrophil-to-lymphocyte ratio, LMR: lymphocyte-to-monocyte ratio, PLR: platelet-lymphocyte ratio, CAR: C-reactive protein albumin ratio.

**Table 3 diagnostics-12-00554-t003:** Predictive factors for IPMN with HGD/INV.

	Univariate		Multivariate	
	OR (95% CI)	*p*-Value	OR (95% CI)	*p*-Value
Age, years	1.02 (0.95–1.00)	0.66		
Sex, male (vs. female)	0.67 (0.26–1.69)	0.39		
IPMN type, Branch duct type (vs. Main duct and Mixed type)	0.94 (0.37–2.26)	0.85		
Location, Head (vs. body and tail)	0.48 (0.19–1.23)	0.13		
Cyst size (mm), <30 (vs. ≥30)	0.99 (0.97–1.02)	0.76		
Number of cyst lesion, Multifocal (vs. Unifocal)	0.91 (0.36–2.3)	0.85		
MPD diameter (mm), ≥10 (vs. <10)	1.29 (0.43–3.85)	0.65		
Jandice, yes (vs. no)	3.68 (0.39–34.5)	0.26		
History of pancreatitis, yes (vs. no)	3.40 (0.66–17.6)	0.15		
Mural nodule, present (vs. absent)	2.3 (0.88–5.99)	0.09		
≥5 Contrast mural nodule, present (vs. absent)	6.18 (2.11–18.1)	<0.01	5.38 (1.71–16.9)	<0.01
CEA (ng/mL), >5 (vs. ≤5)	1.2 (0.43–3.29)	0.73		
CA19-9 (IU/mL), >37 (vs. ≤37)	1.03 (0.28–3.71)	0.97		
NLR, ≥3.27 (<3.27)	0.55 (0.21–1.45)	0.23		
LMR, ≥2.64 (vs. <2.64)	3.75 (0.91–15.5)	0.07	3.10 (0.65–14.9)	0.16
PLR, ≥107 (vs. <107)	0.49 (0.19–1.28)	0.15		
CAR, high (vs. low)	3.28 (1.24–8.69)	0.02	3.84 (1.28–11.5)	0.02

HGD: high grade dysplasia, INV: invasive carcinoma, IPMN: intraductal mucinous neoplasm, MPD: main pancreatic duct, CEA: carcinoembryonic antigen, CA19-9: carbohydrate antigen, NLR: neutrophil-to-lymphocyte ratio, LMR: lymphocyte-to-monocyte ratio, PLR: platelet-lymphocyte ratio, CAR: C-reactive protein albumin ratio, OR: odd ratio, CI: confidence interval.

**Table 4 diagnostics-12-00554-t004:** IPTW logistic odds ratio of IPMN with HGD/INV for CAR^high^.

	Odds Ratio (95% CI)	*p*-Value
Unadjusted	3.28 (1.24–8.69)	0.02
Adjusted for contrast mural nodule, LMR ≥ 2.64	3.84 (1.28–11.5)	0.02
IPTW	4.18 (1.37–12.8)	0.01
IPTW adjusted for contrast mural nodule, LMR ≥ 2.64	4.80 (1.37–16.8)	0.01

LMR: lymphocyte-to-monocyte ratio, IPTW: inverse probability of treatment weighting, CI: Confidence interval.

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
