# Peer review of "Clinical Implication of Preoperative C-Reactive Protein/Albumin Ratio in Malignant Transformation of Intraductal Papillary Mucinous Neoplasm: A Propensity Score Analysis"

_diagnostics, 2022, doi:10.3390/diagnostics12020554_

Round 1

Reviewer 1 Report

would suggest to add a paragraph on potential future clinical implications of this study findings.

Author Response

Responses to the reviewer’s comments:

We appreciate the reviewer’s positive and helpful comments about our paper. The reviewer raised some important points for improvement that we have now addressed, as summarized below. Please note that all changes are yellow highlights in the revised manuscript.

Comment 1:

would suggest to add a paragraph on potential future clinical implications of this study findings.

Response 1:

Thank you for your constructive comments. We agreed with the comment by Reviewer#1. We added the following sentence in the DISCUSSION. Please see the revised manuscript.

We described in the DISCUSSION: "The clinical implications of this study findings are that it can be used as a simple and objective biomarker for malignant transformation of patients with IPMN. Previous reports of NLR and PLR could predict INV but not HGD. However, CAR could predict HGD and INV. Therefore, the high value of CAR will help predict the presence of cancer including HGD and transition to invasive examination such as ERCP and EUS. In addition, this result could predict recurrence in a short period of time before surgery. Since this result can predict the prognosis of patients with IPMN, we believe that it will contribute to study for improving the prognosis in the future." (from page 10, line 276 to 283)

Reviewer 2 Report

This study analyzed the clinical predictive role of the preoperative C-reactive protein/albumin ratio (CAR) for the malignant transformation of intraductal papillary mucinous neoplasm. The data presented supported this possible role of an inflammation-based parameter. The study was done on a cohort of 76 patients in a retrospective design. Even though this fact could be seen as a limitation, the methods and results still revealed CAR's predictive role.

The manuscript is generally well written, presenting the background, methods, results and discussions logically, easy to follow and understand. There are some changes that the authors should be made to improve.

The keywords should not include abbreviated words.

Some editing should be done, correcting the text for unneeded spaces or 

The reference for the histopathological assessment criteria must be included in line 96.

In table 1, there is no need for "yes" in the second column for some parameters.

The images from Figure 2 must be enlarged.

In Table 2, data should be reduced to one value for some rows that were both present or absent, and over and under a cut-off value are presented. It is clear that the 2nd one is the difference. In this way, the Table may be better followed and understood. You can choose the value used in Table 3.

Some checks for the use of the English language should be made: see, for example, lines 247-248 or 259-262. and also in the Conclusions section (line 292-293.

Author Response

Responses to the Reviewer#2 comments:

We appreciate the reviewer’s positive and helpful comments about our paper. The reviewer raised some important points for improvement that we have now addressed, as summarized below. Please note that all changes are yellow highlights in the revised manuscript.

Comment 1:

The keywords should not include abbreviated words.

Response 1:

Thank you for your valuable comment. I revised the manuscript.

Comment 2:

Some editing should be done, correcting the text for unneeded spaces.

Response 2:

Thank you for your valuable comment. I revised the manuscript.

Comment 3:

The reference for the histopathological assessment criteria must be included in line 96.

Response 3:

Thank you for your valuable comment. We added the reference [20].

Comment 4:

In table 1, there is no need for "yes" in the second column for some parameters.

Response 4:

Thank you for your valuable comment. We have removed the "yes" in Table 1. 

Comment 5:

The images from Figure 2 must be enlarged.

Response 5:

Thank you for your valuable comment. We have enlarged Figure 2. 

Comment 6:

In Table 2, data should be reduced to one value for some rows that were both present or absent, and over and under a cut-off value are presented. It is clear that the 2nd one is the difference. In this way, the Table may be better followed and understood. You can choose the value used in Table 3.

Response 6:

Thank you for your valuable comment. We have removed the Absent column. Regarding Mural nodule, We changed the display to include IQR as a continuous variable. As for > 5mm contrast mural nodule, we have not changed them because they are categories. 

Comment 7:

Some checks for the use of the English language should be made: see, for example, lines 247-248 or 259-262. and also in the Conclusions section (line 292-293.

Response 7:

Thank you for your valuable comment. This manuscript has been edited by the native speaker of editage. 

Reviewer 3 Report

I’ve carefully read the manuscript entitled “Clinical implication of preoperative C-reactive protein/albumin ratio in malignant transformation of intraductal papillary mucinous neoplasm: a propensity score analysis”, which describes the diagnostic performance of CAR in predicting malignant transformation of IPMNs.

Background on the topic is well delineated, introducing the author to challenges of current management decisions in IPMN.

Methodology is clearly defined.

Results are presented nicely with corresponding tables and graphs.  

Beyond their value as negative/positive acute phase proteins (which are impaired in cancer patients), the authors should also discuss other potential confounding factors such as liver disease or other pathology which alter protein levels. Also, my suggestion is to include an analysis of diagnostic accuracy of CAR vs. management according to current guidelines.

Author Response

Responses to the Reviewer#3 comments:

We appreciate the reviewer’s positive and helpful comments about our paper. The reviewer raised some important points for improvement that we have now addressed, as summarized below. Please note that all changes are yellow highlights in the revised manuscript.

Comment 1:

Beyond their value as negative/positive acute phase proteins (which are impaired in cancer patients), the authors should also discuss other potential confounding factors such as liver disease or other pathology which alter protein levels. Also, my suggestion is to include an analysis of diagnostic accuracy of CAR vs. management according to current guidelines.

Response 1:

Thank you for your constructive comment. There was one case of liver cirrhosis in this study. It has been adjusted by IPTW along with other comorbidities and is reflected in the results in Table 4. An analysis of diagnostic accuracy of CAR vs. management according to current guidelines was shown in Table 3. The items other than > 5mm contrast mural nodule were no significant difference in the univariate analysis, and the diagnostic accuracy was inferior to CAR.

Reviewer 4 Report

Dear Authors,

I have read with great interest your research article titled: Clinical implication of preoperative C-reactive protein/albumin ratio in malignant transformation of intraductal papillary mucinous neoplasm: a propensity score analysis. Below you can find suggestions to improve the overall quality of your manuscript.

Minor points:

Please check once again the entire manuscript for typos: du e, in vasive

Lines 94-96, 2.5 Histopathological assessment - a reference is needed.

Major points:

The quality of Figure 4 (A) and (B) must be improved because it is overexpressed.

The relatively small number of patients included in the study (n = 76).

What novelty brings this paper compared to the existing literature, in spite of articles with a similar design?

Kind regards,

The Reviewer

Author Response

Responses to the Reviewer#4 comments:

We appreciate the reviewer’s positive and helpful comments about our paper. The reviewer raised some important points for improvement that we have now addressed, as summarized below. Please note that all changes are yellow highlights in the revised manuscript.

Comment 1:

Please check once again the entire manuscript for typos: du e, in vasive

Response 1:

Thank you for your constructive comment. We revised the manuscript.

Comment 2:

Lines 94-96, 2.5 Histopathological assessment - a reference is needed.

Response 2:

Thank you for your constructive comment. We added the reference [20].

Comment 3:

The quality of Figure 4 (A) and (B) must be improved because it is overexpressed.
The relatively small number of patients included in the study (n = 76).

Response 3:

Thank you for your constructive comment. Since this is a retrospective study, the final observation period will be different for each patient. Therefore, it is a characteristic of the Kaplan-Meier curve that censoring occurs in each patient even if no event (death, recurrence) occurs. The same figure is obtained for R and SPSS. This is the result of this study and cannot be revised.
I hope you will understand this.

Comment 4:

What novelty brings this paper compared to the existing literature, in spite of articles with a similar design?

Response 4:

Thank you for your constructive comment. Previous reports of NLR and PLR could predict INV but not HGD. However, CAR could predict HGD and INV. Therefore, the high value of CAR will help predict the presence of cancer including HGD and transition to invasive examination such as ERCP and EUS. In addition, this result could predict recurrence in a short period of time before surgery. The clinical implications of this study findings are that it can be used as a simple and objective biomarker for malignant transformation of patients with IPMN. 
We added the sentence in the DISCUSSION. Please see the revised manuscript. (Page 10, line 276 to 283)

Reviewer 5 Report

The study “Clinical implication of preoperative C-reactive protein/albumin ratio in malignant transformation of intraductal papillary mucinous neoplasm: a propensity score analysis” by Maruyama et al. performs a retrospective study in order to use C-reactive protein albumin ratio (CAR) as a predictive marker of malignant transformation of intraductal papillary mucinous neoplasms (IPMNs) due to the difficulty of knowing its benign or malignant nature with imaging techniques. The study is original in terms of the marker used and can offer new horizons in the early diagnosis of the development of this subtype of cancer.

However, I have some comments to make:

  • Are the data in table 1 for NLR, PLR, LMR and CAR the same as those represented in the diagrams in figure 2?
  • On the other hand, how is it explained that the NLR and PLR data do not coincide with what was published with other authors in IPMN?
  • Are 76 patients enough to draw these conclusions?

Author Response

Responses to the Reviewer#5 comments:

We appreciate the reviewer’s positive and helpful comments about our paper. The reviewer raised some important points for improvement that we have now addressed, as summarized below. Please note that all changes are yellow highlights in the revised manuscript.

Comment 1:

Are the data in table 1 for NLR, PLR, LMR and CAR the same as those represented in the diagrams in figure 2?

Response 1:

Thank you for your constructive comment. These are same data. I hope you will understand this.

Comment 2:

On the other hand, how is it explained that the NLR and PLR data do not coincide with what was published with other authors in IPMN?

Response 2:

Thank you for your constructive comment. Previous reports of NLR and PLR could predict INV but not HGD. However, CAR could predict HGD and INV. We could predict for those with mild systemic inflammation and good nutritional status using CAR for evaluation. In this study, we believe that the use of high-sensitivity CRP reflects minor changes that do not affect NLR or PLR. We described in the DISCUSSION. Please see the revised manuscript. (Page 10, line 276 to 284)

Comment 3:

Are 76 patients enough to draw these conclusions?

Response 3:

Thank you for your valuable comment. We agreed with the comment by Reviewer#5. We described the "limitation" in the DISCUSSION. I hope you will understand this.

Round 2

Reviewer 4 Report

Dear Authors,

Congratulations on the hard work reviewing your manuscript. The quality of the manuscript has been significantly improved.

The Reviewer

Author Response

Thank you for your comments.

Reviewer 5 Report

The authors have done a good job and have improved the manuscript following the instructions of the reviewers.

Author Response

Thank you for your comments.